# ProtoGate: Prototype-based Neural Networks with Local Feature Selection for Tabular Biomedical Data

**Xiangjian Jiang** [1]    **Andrei Margeloiu** [1]    **Nikola Simidjievski** [1]    **Mateja Jamnik** [1]

## Abstract

Tabular biomedical data poses challenges in machine learning because it is often high-dimensional and typically low-sample-size. Previous research has attempted to address these challenges via feature selection approaches, which can lead to unstable performance and insufficient interpretability on real-world data. This suggests that current methods lack appropriate inductive biases that capture informative patterns in different samples. In this paper, we propose ProtoGate, a local feature selection method that introduces an inductive bias by attending to the clustering characteristic of biomedical data. ProtoGate selects features in a global-to-local manner and leverages them to produce explainable predictions via an interpretable prototype-based model. We conduct comprehensive experiments to evaluate the performance of ProtoGate on synthetic and real-world datasets. Our results show that exploiting the homogeneous and heterogeneous patterns in the data can improve prediction accuracy while prototypes imbue interpretability.

## 1. Introduction

In biomedical research, tabular data is frequently collected (Baxevanis et al., 2020; Lesk, 2019; Polanski & Kimmel, 2007) for a wide range of applications such as detecting marker genes (Hsu et al., 2003), identifying cancer subtypes (Hsu et al., 2003), and performing survival analysis (Yang et al., 2022; Fan et al., 2022). Clinical trials, whilst collecting large amounts of high-dimensional data using modern high-throughput sequencing technologies, often consider a small number of patients due to practical reasons (Levin et al., 2022). The resulting tabular datasets are

thus often high-dimensional and typically low-sample-size (HDLSS). Moreover, given the inherent heterogeneity of biomedical data, important features often vary from sample to sample – even in the same dataset (Yang et al., 2022; Yoon et al., 2018). Such scenarios have proven challenging for current machine learning approaches, including deep tabular models (Liu et al., 2017; Shwartz-Ziv & Armon, 2022; Mamoshina et al., 2016; Yang et al., 2022; Aoshima et al., 2018; Margeloiu et al., 2023).

Previous methods (Remeseiro & Bolon-Canedo, 2019; Chen et al., 2018; Yoon et al., 2018; Yang et al., 2022) have attempted to address such challenges by performing local feature selection: rather than selecting a general set of important features across all samples, local feature selection methods select specific subsets of features for each sample and these subsets may vary from sample to sample. However, existing methods have three limitations: (i) In many real-world tasks, even simple models – such as an MLP or Lasso – can outperform many existing methods (Yang et al., 2022). One reason is the accuracy of current methods can be substantially lower for some classes than other classes, and we illustrate this in Figure 1. (ii) These methods commonly comprise a trainable feature selector to select features and a trainable predictor to make predictions with these features, which can be susceptible to the co-adaptation problem (Jethani et al., 2021; Adebayo et al., 2018; Hooker et al., 2019). Because the two components are jointly trained, the predictor can fit the selected features to achieve high accuracy even when these features do not reflect the real data distribution (Jethani et al., 2021). Consequently, the prediction accuracy is inconsistent with the quality of selected features. For instance, L2X (Chen et al., 2018) achieves 96% accuracy in digit classification on MNIST by using only one pixel as input (Jethani et al., 2021). (iii) Current methods (Yoon et al., 2018; Jethani et al., 2021; Yang et al., 2022; Chen et al., 2018) cannot provide explainable predictions because they mainly use an MLP-based predictor. This lack of explainability is a major concern in high-stake applications such as medicine (Holzinger et al., 2019; London, 2019; Amann et al., 2020; Reddy, 2022; Tjoa & Guan, 2020).

We hypothesise that these local feature selection methods exhibit subpar performance for two reasons:

---

[1]Department of Computer Science and Technology, University of Cambridge, UK. Correspondence to: Xiangjian Jiang <xj265@cam.ac.uk>.

*Workshop on Interpretable ML in Healthcare at International Conference on Machine Learning (ICML)*, Honolulu, Hawaii, USA. 2023. Copyright 2023 by the author(s).

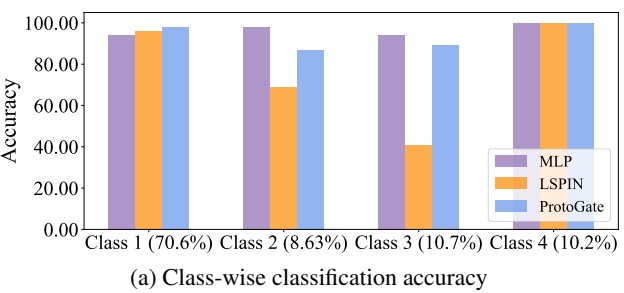
(a) Class-wise classification accuracy

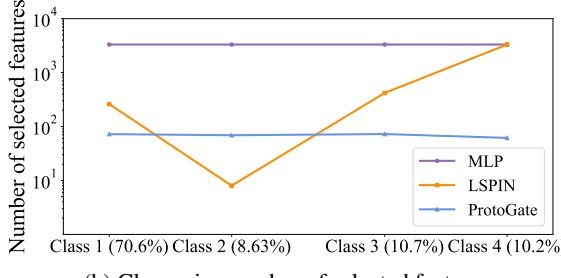
(b) Class-wise number of selected features

*Figure 1.* Illustration of the unstable performance of LSPIN (Yang et al., 2022) on the lung dataset. **(a)** The class-wise classification accuracy, with class distribution in the parentheses. **(b)** The mean number of selected features, displayed on a logarithmic scale. MLP and ProtoGate achieve stable performance, while LSPIN has a large variance in the accuracy and number of selected features across classes.

- **Lack of appropriate inductive biases:** These methods mainly make predictions using MLPs, although, in the biomedical domain, the clustering assumption (which states similar samples should belong to the same class (Chapelle et al., 2006)) has been shown effective (Kolodner, 1992; Li et al., 2018; Bichindaritz & Marling, 2006; Bichindaritz, 2008; Lu et al., 2021). Based on the clustering assumption, the prototype-based models can perform well on tabular data by classifying the new instances according to their similarities to the existing prototypes. For instance, a simple prototype-based model, such as $k$-means, can outperform complex neural networks with an accurate pre-trained feature selector (Yang et al., 2022).

- **Lack of consideration on the homogeneity between samples:** Existing local feature selection models tend to overly emphasize heterogeneity, often neglecting that different samples might share some informative features. The high accuracy of global feature selection models on real-world datasets (Margeloiu et al., 2023) suggests that informative features can indeed be shared across samples.

Therefore, we believe that effective local feature selection methods should be able to identify the clustering characteristics and homogeneous/heterogeneous feature patterns across samples, provided the data supports their existence.

In this paper, we aim to address the challenges of suboptimal performance and the opaqueness of local feature selection methods applied to tabular biomedical data. We propose ProtoGate, a novel method which performs local feature selection and makes accurate and explainable predictions in the HDLSS regime.

Firstly, ProtoGate uses a prototype-based predictor without learnable parameters – namely Differentiable K-Nearest Neighbors (DKNN) (Grover et al., 2019) – which enables explainable predictions. The prototype-based predictor confers ProtoGate two important features: (i) an inductive bias aligned with the clustering assumption in biomedical data;

and (ii) consistent evaluations of the quality of selected features throughout the training process, eliminating the possibility of co-adaptation from joint training. Secondly, ProtoGate performs feature selection in a global-to-local manner with an $\ell_1$-regularised gating network. The global-to-local design helps ProtoGate consider the homogeneous and heterogeneous patterns across multiple samples.

Our contributions can be summarised as follows:

1. We propose ProtoGate, a novel method which addresses the challenge of high-dimensional and low-sample-size (HDLSS) biomedical data by achieving local feature selection and explainable predictions with a global-to-local feature selector and a prototype-based classifier.

2. We show that ProtoGate generally outperforms 12 benchmark methods on seven real-world biomedical datasets (Section 4.1) while selecting fewer features (Section 4.2), paving the path to more robust and interpretable local feature selection models.

3. We demonstrate that ProtoGate effectively handles the co-adaptation problem via a prototype-based predictor without learnable parameters by evaluating its performance on three synthetic datasets (Section 4.3).

## 2. Related Work

**Feature Selection Methods** Feature selection is a common technique for improving the accuracy and interpretability of machine learning models on HDLSS datasets. An extensive line of work selects features globally with Lasso-based regularisation (Tibshirani, 1996; Feng & Simon, 2017; Yamada et al., 2014; 2018; Climente-González et al., 2019) or specialised layers in neural networks (Margeloiu et al., 2023; Singh et al., 2020; Balın et al., 2019; Lemhadri et al., 2021; Yamada et al., 2020). However, the global feature selection ignores the heterogeneous nature of biomedical data, leading to insufficient interpretability (Yang et al., 2022).

Prior studies attend to the heterogeneity between samples

by designing post-hoc local explanation methods to explain the instance-wise feature importance for a pre-trained predictor (Jethani et al., 2021; Ribeiro et al., 2016; Lundberg & Lee, 2017; Shrikumar et al., 2017; Simonyan et al., 2013; Lundberg et al., 2018; Bach et al., 2015). These methods are limited because the post-hoc analysis on feature importance does not improve the performance of pre-trained predictors.

Recent work proposes to select instance-wise features for making predictions (Yoon et al., 2018; Yang et al., 2022; Chen et al., 2018; Arik & Pfister, 2021; Yoshikawa & Iwata, 2022). L2X uses mutual information for instance-wise feature selection with Concrete distribution, but it requires specifying the number of selected features (Chen et al., 2018). INVASE addresses such limitation by modelling each feature's mask/gate value with independent Bernoulli distributions (Yoon et al., 2018). However, both methods utilise computationally expensive gradient estimators: RE-INFORCE (Williams, 1992) or REBAR (Tucker et al., 2017). Similar to STG (Yamada et al., 2020), LSPIN/LLSPIN reformalises the mask/gate value with injected Gaussian noise and extends Localized Lasso (Yamada et al., 2017) with a gating network that can select similar features for similar samples (Yang et al., 2022). However, the poor performance of a vanilla KNN on real-world datasets (Table 1) demonstrates that the similarity in the initial high-dimensional feature space is inaccurate because a large proportion of features can be noise for the prediction. And these methods mainly employ unexplainable MLPs for making predictions, which can be unsuitable for tabular biomedical data (Shwartz-Ziv & Armon, 2022).

In contrast, ProtoGate measures the similarity across samples within an intrinsically interpretable DKNN predictor. The predictor takes the samples after feature selection as input, and thus the similarity is measured in a feature space with reduced dimensions.

**Prototype-based Machine Learning** Prototype-based models (Biehl et al., 2016) in machine learning are closely related to metric learning (Goldberger et al., 2004) and case-based reasoning (Kolodner, 1992). They are built upon the clustering assumption and aim to represent data through prototypical exemplars (e.g. KNN (Fix, 1985)) or a set of prototypical centroids (e.g. $k$-means (Ball & Hall, 1965)) that capture the fundamental characteristics of the data. These core ideas parallel similar concepts from cognitive psychology and neurosciences (Biehl et al., 2016). Therefore, prototype-based models have gained attention for their potential to improve the performance and interpretability of machine learning approaches, particularly in the context of the biomedical field (Kolodner, 1992; Li et al., 2018; Bichindaritz & Marling, 2006; Bichindaritz, 2008; Lu et al., 2021). Therefore, ProtoGate employs a prototype-based predictor for explainable predictions on biomedical data.

**Co-adaptation Problem** In feature selection, co-adaptation refers to the situation where the model encodes predictions into the feature selection, leading to high accuracy with features that do not reflect the real data distributions (Jethani et al., 2021; Adebayo et al., 2018; Hooker et al., 2019; Samek et al., 2016). REAL-X proves that co-adaptation can happen in models with jointly trained feature selectors and predictors (Jethani et al., 2021), and addresses this problem by decoupling the training objectives of the feature selector and predictor. However, it is important to note that REAL-X can only provide a post-hoc analysis of feature importance for individual samples, which does not address the performance gap arising from HDLSS data.

In ProtoGate, we propose to address the co-adaptation problem with DKNN, a prototype-based predictor without learnable parameters. Therefore, the DKNN predictor cannot adapt to the feature selector, eliminating the possibility of co-adaptation from joint training.

## 3. Method

### 3.1. Problem Setup

We consider the classification task on tabular biomedical data with $\mathcal{Y}$ classes. Let $X := [\boldsymbol{x}^{(1)}, \ldots, \boldsymbol{x}^{(N)}]^\top \in \mathbb{R}^{N \times D}$ be the data matrix consisting of $N$ samples $\boldsymbol{x}^{(i)} \in \mathbb{R}^D$ with $D$ features, and let $Y := [y^{(1)}, \ldots, y^{(N)}] \in \mathbb{R}^N$ be the corresponding labels. We denote $x_d^{(i)}$ as the $d$-th feature of the $i$-th sample. To simplify the notation, we assume all samples in $X$ are used for training.

A common local feature selection model contains two components: (i) an instance-wise feature selector $S_{\mathbf{W}} : \mathbb{R}^D \to \{0, 1\}^D$ that takes as input a sample $\boldsymbol{x}^{(i)}$ and generates a mask/gate $\boldsymbol{s}^{(i)} \in [0, 1]^D$ for its features, and (ii) a predictor model $F_\theta : \mathbb{R}^D \to \mathcal{Y}$ which takes as input both the sample $\boldsymbol{x}^{(i)}$ and the mask $\boldsymbol{s}^{(i)}$ for prediction:

$$\hat{y}^{(i)} = F_\theta \left( S_{\mathbf{W}}(\boldsymbol{x}^{(i)}), \boldsymbol{x}^{(i)} \right) = F_\theta(\boldsymbol{x}^{(i)} \odot \boldsymbol{s}^{(i)}) \quad (1)$$

where $\hat{y}^{(i)}$ is the predicted label and $\odot$ is the element-wise multiplication. Here, we define the $d$-th feature is selected if and only if the mask value is positive ($s_d^{(i)} > 0$).

### 3.2. Rationales for Model Design

We propose ProtoGate as a method for selecting instance-wise features with inductive biases from the prototype-based model, as shown in Figure 2. And the pseudocode for model training is summarised in Algorithm 1.

Firstly, ProtoGate selects instance-wise features with a global-to-local feature selector, which is an $\ell_1$-regularised gating network. This design allows the feature selector to consider the homogeneous and heterogeneous feature patterns across samples.

Secondly, ProtoGate leverages DKNN as the predictor, a prototype-based model. The differentiability allows DKNN to encode the clustering assumption into feature selection when trained in tandem with the feature selector. Furthermore, the DKNN predictor is inherently interpretable. And it can avoid the co-adaptation problem because it has no learnable parameters to fit the selected features.

### 3.3. Global-to-local Feature Selection (Figure 2 (A))

The global-to-local feature selector $S_{\mathbf{W}} : \mathbb{R}^D \to [0,1]^D$ contains a neural network that maps feature values $\boldsymbol{x}^{(i)}$ into mask values $\boldsymbol{s}^{(i)}$. The feature selector attends to the homogeneity between samples via applying $\ell_1$-regularisation on $\mathbf{W}^{[1]}$, the weights of the first layer. Intuitively, the regularisation can lead to sparse weights in the first layer, which implicitly selects features globally for all samples. The output $\boldsymbol{\mu}^{(i)}$ from the last layer is thresholded to obtain instance-wise mask values by

$$s_d^{(i)} = \max(0, \min(1, \mu_d^{(i)} + \epsilon_d^{(i)})) \tag{2}$$

where $\epsilon_d^{(i)}$ is the injected noise sampled from Gaussion distribution $\mathcal{N}(0, \sigma^2)$. The standard deviation $\sigma$ is fixed during training, and it is removed during the inference time for deterministic mask values.

With the injected noise, $\boldsymbol{s}^{(i)}$ can be re-formalised as random vectors whose parameters $\boldsymbol{\mu}^{(i)}$ are predicted by a neural network. Therefore, the sparsity regularisation on mask values can be computed by

$$R(\mathbf{W}^{[1]}, \boldsymbol{s}^{(i)}, \lambda_g, \lambda_l) = \lambda_g ||\mathbf{W}^{[1]}||_1 + \mathbb{E}\left[\lambda_l ||\boldsymbol{s}^{(i)}||_0\right]$$
$$= \lambda_g ||\mathbf{W}^{[1]}||_1 + \lambda_l \sum_{d=1}^{D}\left(\frac{1}{2} - \frac{1}{2}\mathrm{erf}(-\frac{\mu_d^{(i)}}{\sqrt{2}\sigma})\right) \tag{3}$$

where $(\lambda_g, \lambda_l)$ is a pair of hyper-parameters to balance the effects of global and local feature selection, and $\mathrm{erf}(\cdot)$ is the Gauss error function. The full derivations are available in Appendix A.2. By considering the interplay between $\lambda_g$ and $\lambda_l$, ProtoGate can perform local feature selection with both homogeneity and heterogeneity across samples considered.

### 3.4. Prototype-based Prediction (Figure 2 (C))

The prototype-based predictor $F : \mathbb{R}^D \to \mathcal{Y}$ is a DKNN model. The DKNN predictor first constructs a prototype base $\mathcal{B}$ with training samples. After masking the training samples with the mask generated by $S_{\mathbf{W}}(X)$, DKNN retains the masked samples and their labels as prototypes in the base $\mathcal{B} := \{(S_{\mathbf{W}}(\boldsymbol{x}^{(i)}) \odot \boldsymbol{x}^{(i)}, y^{(i)})\}_{i=1}^{N}$. With the acquired prototypes, the predictor can classify a query sample $\boldsymbol{x}_{\mathrm{query}} \in X$ by retrieving the base $\mathcal{B}$. The predictor sorts the

prototypes by their similarities to the masked query sample with NeuralSort (Grover et al., 2019), a differentiable relaxed sorting operator. Note that ProtoGate computes the Euclidean distance between samples as the similarity evaluation metric. According to the sorting results, the predictor uses the majority class of the $K$ closest prototypes as the predicted label $\hat{y}_{\mathrm{query}}$. For each query sample, the loss of prototype-based classification is defined as:

$$\ell_{\mathrm{pred}}(P_{\mathcal{B}}^{\mathrm{query}}, \boldsymbol{x}_{\mathrm{query}}, y_{\mathrm{query}}) = K - \ell_{\mathrm{NeuralSort}}$$
$$= K - \frac{1}{K}\sum_{j=1}^{K}\sum_{i=1}^{N} \mathbb{1}\left(y^{(i)} = y_{\mathrm{query}}\right) P_{\mathcal{B}}^{\mathrm{query}}[i,j] \tag{4}$$

where $P_{\mathcal{B}}^{\mathrm{query}} \in \mathbb{R}^{N \times N}$ is the relaxed permutation matrix and $\mathbb{1}(\cdot)$ is the indicator function. In the permutation matrix, $P_{\mathcal{B}}^{\mathrm{query}}[i,j]$ denotes the possibility that the $i$-th prototype is the $j$-th closest to query sample $\boldsymbol{x}_{\mathrm{query}}$ under NeuralSort. Among the $K$ nearest prototypes, Equation 4 estimates the number of prototypes that have different labels to $\boldsymbol{x}_{\mathrm{query}}$.

Because the feature selector is learnable, the predicted masks can change over the training time, and thus the prototypes in base $\mathcal{B}$ are changing before the model converges. After training, the prototype base $\mathcal{B}$ is fixed, and query samples are from unseen test data.

### 3.5. Training Loss

The training loss in ProtoGate consists of two components: the classification loss and the sparsity regularisation term. Equation 5 shows the formulation of the total loss:

$$L = \frac{1}{N}\sum_{i=1}^{N}\left(\ell_{\mathrm{pred}}(P_{\mathcal{B}}^{(i)}, \boldsymbol{x}^{(i)}, y^{(i)}) + R(\mathbf{W}^{[1]}, \boldsymbol{s}^{(i)}, \lambda_g, \lambda_l)\right). \tag{5}$$

Because the loss function is fully differentiable, the global-to-local feature selector and the prototype-based predictor can be trained in tandem. The whole model can be optimised with standard gradient-based approaches, such as stochastic gradient descent. We did not observe optimisation issues when training over 3,000 models (Appendix B.4).

## 4. Experiments

We now evaluate ProtoGate on both synthetic and real-world datasets to substantiate the model design choices. Firstly, we compare ProtoGate against 12 benchmark methods on real-world classification tasks (Section 4.1 and Section 4.2). Secondly, we investigate the impact of the prototype-based predictor by replacing it with a linear or MLP-based prediction head (Appendix C.1) and adjusting the number of nearest neighbours (Appendix C.2). Thirdly, we analyse the co-adaptation problem by considering the performance misalignment between feature selection and classification

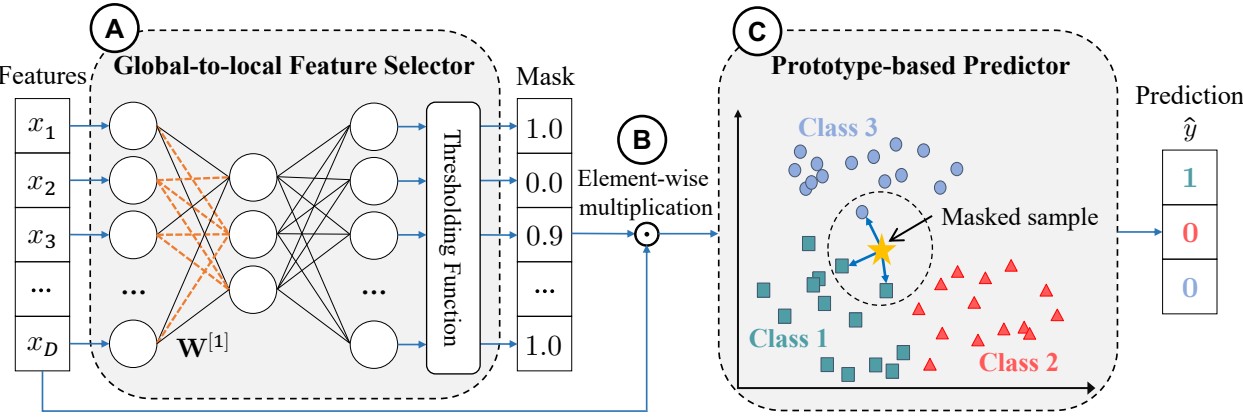

*Figure 2.* The architecture of ProtoGate. **(A)** Given a sample $x \in \mathbb{R}^D$, the global-to-local feature selector performs global feature selection in the first layer. The orange dashed lines denote sparsified weights in $\mathbf{W}^{[1]}$ under $\ell_1$ regularisation. The neural network then computes the instance-wise mask values $\{s_d\}_{d=1}^D \in [0,1]^D$ with a thresholding function. **(B)** The mask is applied to the sample for local feature selection by element-wise multiplication. **(C)** The prototype-based predictor classifies $x$ by retrieving the $K$ nearest neighbours to the masked sample in base $\mathcal{B}$. The majority class of neighbours is used as the predicted label $\hat{y}$.

*Table 1.* Evaluation predictive performance of ProtoGate with 12 benchmark methods on seven real-world tabular biomedical datasets. We report the mean $\pm$ standard deviation of test balanced accuracy (averaged across 25 runs) and average accuracy rank across datasets. A smaller rank implies higher accuracy. Note that INVASE failed to converge on 3 datasets, and we computed its rank with the averaged balanced accuracy of other methods on corresponding datasets. We highlight the **First**, **Second** and **Third** ranking accuracy for each dataset. ProtoGate consistently ranks Top-3 across datasets and achieves the best overall performance.

| Methods | lung | meta-dr | meta-pam | prostate | tcga-2y | toxicity | colon | Avg. Rank |
|---|---|---|---|---|---|---|---|---|
| LightGBM | 93.42 ± 5.91 | 58.23 ± 8.56 | 94.98 ± 5.19 | **91.38 ± 5.71** | 57.09 ± 7.87 | 81.98 ± 6.25 | 76.60 ± 11.67 | 5.71 |
| RF | 91.73 ± 6.61 | 51.48 ± 3.41 | 88.73 ± 6.24 | 90.38 ± 7.31 | 58.70 ± 6.84 | 79.78 ± 7.10 | **80.05 ± 10.37** | 7.14 |
| KNN | 91.06 ± 7.92 | 54.64 ± 7.95 | 82.79 ± 9.20 | 78.78 ± 6.71 | **58.83 ± 7.07** | 83.86 ± 12.03 | 77.33 ± 5.41 | 8.00 |
| Lasso | **94.47 ± 3.17** | **58.58 ± 9.04** | 95.15 ± 2.83 | **91.18 ± 6.39** | 56.99 ± 6.26 | **91.86 ± 5.27** | 79.40 ± 8.50 | **4.29** |
| MLP | **95.81 ± 2.69** | 54.68 ± 9.63 | **95.71 ± 2.59** | 87.22 ± 7.41 | 55.32 ± 7.24 | **93.54 ± 4.28** | 80.00 ± 8.70 | **5.14** |
| STG | 93.30 ± 6.28 | 58.15 ± 8.67 | 76.13 ± 8.19 | 89.38 ± 5.85 | 57.04 ± 5.76 | 87.95 ± 5.01 | 79.55 ± 10.53 | 6.29 |
| TabNet | 77.65 ± 11.56 | 49.18 ± 15.02 | 82.66 ± 7.81 | 65.66 ± 9.03 | 51.58 ± 8.26 | 40.06 ± 12.23 | 56.75 ± 7.31 | 12.00 |
| L2X | 50.02 ± 8.30 | 52.54 ± 13.75 | 62.64 ± 13.69 | 61.78 ± 6.29 | 52.30 ± 9.11 | 31.72 ± 13.48 | 57.60 ± 14.26 | 12.43 |
| INVASE | 91.22 ± 6.16 | — | 91.70 ± 6.84 | — | 55.98 ± 6.45 | 80.04 ± 6.60 | — | 9.00 |
| REAL-X | 93.27 ± 4.32 | **60.01 ± 7.12** | 95.59 ± 3.04 | 86.75 ± 6.68 | **59.30 ± 7.49** | 90.79 ± 4.75 | 76.75 ± 12.21 | **5.14** |
| LLSPIN | 70.10 ± 12.31 | 56.77 ± 9.65 | 95.50 ± 3.60 | 88.71 ± 5.98 | 57.87 ± 6.02 | 81.67 ± 9.01 | 79.35 ± 7.74 | 7.14 |
| LSPIN | 76.92 ± 9.38 | 53.98 ± 8.00 | **97.18 ± 3.16** | 87.75 ± 6.74 | 55.95 ± 7.45 | 83.47 ± 8.59 | **81.30 ± 7.97** | 6.71 |
| **ProtoGate** | **93.44 ± 6.37** | **60.43 ± 7.61** | **95.96 ± 3.93** | **90.58 ± 5.64** | **61.18 ± 6.47** | **92.34 ± 5.67** | **81.10 ± 12.14** | **2.00** |

on the synthetic datasets (Section 4.3). We also provide the comparison of training time in Appendix D.

**Real-world datasets.** Following (Margeloiu et al., 2023), we utilise seven HDLSS tabular biomedical datasets. The datasets contain $2000 - 5966$ features with $62 - 197$ samples of $2 - 4$ different classes. We are interested in datasets with much fewer samples than LSPIN (Yang et al., 2022), which uses $\sim 1,500$ samples. Full descriptions of the real-world datasets are available in Appendix B.1.

**Experimental setup.** For each dataset, we perform 5-fold cross-validation on 5 different splits, summing up to 25 runs per model. We obtain the validation set by randomly selecting 10% of training data. For each benchmark model, the training loss is a weighted loss, and we perform a hyper-parameter search for model selection on the validation set.

Full details about the reproducibility and hyper-parameter tuning are available in Appendix B.5.

**Evaluation metrics.** We report the results averaged over 25 runs on test samples. (i) For classification, we measure the performance by the mean $\pm$ standard deviation test balanced accuracy. (ii) Note that the proportion of selected features varies across samples for local feature selection methods. Therefore, we measure the sparsity of feature selection by the mean $\pm$ standard deviation proportion of selected features across samples. (iii) To distinguish between "similar number of selected features" and "similar selected features", we introduce a new metric: degree of

local sparsity $\mathcal{Q}$, which is computed by

$$\mathcal{Q} = \frac{1}{D \cdot N} \sum_{j=1}^{N} \text{card} \left( \bigcup_{i=1}^{N} \text{nonzero}(\boldsymbol{s}^{(i)}) - \text{nonzero}(\boldsymbol{s}^{(j)}) \right)$$

(6)

where card$(\cdot)$ returns the cardinality of a set and nonzero$(\cdot)$ returns the indices of non-zero elements in a vector. $\mathcal{Q}$ measures the difference between the union set of selected features for all samples and the selected features for a specific sample. Intuitively, a non-zero $\mathcal{Q}$ denotes selected features are different across samples, and thus the feature selection is local. For global feature selection, the degree of local sparsity is zero ($\mathcal{Q} \equiv 0$).

**ProtoGate implementation.** The global-to-local feature selector is flexible on the number of hidden layers, and we implement it as a three-layer feed-forward neural network. The numbers of neurons in the input and output layers are the same as the number of features of the input data, and the hidden layer has 100/200 neurons. The feature selector has batch normalisation and $tanh$ activation for all layers. We train the models with a batch size of 64 and utilise an SGD optimiser with a weight decay of $1e - 4$. The number of nearest neighbours $K$ is searched in $\{1, 2, 3, 4, 5\}$. The global sparsity hyper-parameter $\lambda_g$ is searched in $\{1e - 4, 2e - 4, 3e - 4, 4e - 4, 6e - 4\}$, and the local sparsity hyper-parameter $\lambda_l$ is set as $1e - 3$. Full implementation details are available in B.5.

**Benchmark methods.** We evaluate the classification accuracy of ProtoGate and compare it with several benchmark models, including global feature selection models (Light-GBM (Ke et al., 2017), Random Forest (RF) (Breiman, 2001), Lasso (Tibshirani, 1996) and STG (Yamada et al., 2020)) and local feature selection models (Tab-Net (Arik & Pfister, 2021), L2X (Chen et al., 2018), IN-VASE (Yoon et al., 2018), REAL-X (Jethani et al., 2021) and LSPIN/LLSPIN (Yang et al., 2022)). Additionally, we also compare ProtoGate with some standard models, including KNN (Fix, 1985) and MLP.

### 4.1. Classification Performance

Table 1 shows that ProtoGate consistently achieves better than or comparable balanced accuracy to the benchmark models. We compute the average rank across different datasets, and ProtoGate ranks first, followed by Lasso. ProtoGate outperforms all other local feature selection models by a clear margin. We also find that the existing local feature selection methods cannot outperform even the simple linear Lasso or vanilla MLPs on HDLSS datasets. Note that REAL-X achieves comparable performance as the vanilla MLP model because it trains an MLP-based predictor with all features.

The stable and competitive performance of ProtoGate shows

the suitability of the clustering assumption in the biomedical field. Moreover, ProtoGate intrinsically provides explanations for the predictions by explicitly pointing out the $K$ nearest prototypes, while other local feature selection methods can be unexplainable with MLP-based predictors. Poor performance of the vanilla KNN model also demonstrates that a large proportion of features can be irrelevant to the predictions, and thus the similarity in the high-dimensional feature space can introduce noise to feature selection, which can be one reason for the failure of LSPIN and LLSPIN.

In most HDLSS cases, ProtoGate consistently outperforms both Lasso and MLP. ProtoGate could have slightly lower accuracy on some datasets, such as the toxicity dataset. We attribute this to the limited expressivity of prototype-based models compared to connectionist models (Lu et al., 2017). As mentioned in (Margeloiu et al., 2023; Yang et al., 2022), Lasso and MLP can outperform other feature selection models when they are well-regularised on some datasets, such as the toxicity dataset. Compared with well-regularised Lasso and MLP, the prototype-based predictor could have limited expressivity, but ProtoGate still has higher overall accuracy.

### 4.2. Feature Selection Performance

We compare ProtoGate against global feature selection methods (RF, Lasso and STG) and local feature selection methods (L2X, LSPIN and LLSPIN). We plot the mean $\pm$ standard deviation of the proportion of selected features across samples in Figure 4 and heatmaps of mask values in Figure 5. The numerical results and full visualisations of selected features are available in Appendix E.

**Number of selected features.** Figure 4 shows that ProtoGate consistently selects fewer features per sample than other benchmark methods, except L2X. Because the performance of L2X is the worst among the 12 benchmark models, we argue that the L2X model does not perform better than ProtoGate on feature selection, although it has the fewest selected features. Compared with the rest local feature selection methods, ProtoGate has smaller standard deviations in the proportion of selected features across test samples. Note that this does not mean ProtoGate selects features globally because the degree of local sparsity is positive (Figure 3(b)).

The sparse feature selection results from ProtoGate demonstrate the effectiveness of global information in feature selection, and the global-to-local process helps ProtoGate attend to both homogeneity and heterogeneity across samples.

**Degree of local sparsity.** We further examine how different hyper-parameter values of global sparsity $\lambda_g$ impact the feature selection behaviour.

Figure 3(a) shows that increasing $\lambda_g$ can lead to a lower degree of local sparsity. On most datasets, ProtoGate starts to perform global feature selection with $\lambda_g \geq 1e - 3$. At

*Table 2.* Evaluation comparison of ProtoGate and nine benchmark methods on three synthetic datasets. We report the F1 score of selected features (F1$_{\text{selec}}$) and the balanced accuracy for prediction (ACC$_{\text{pred}}$) on test samples. 'Diff.' refers to the difference between the ranks of F1$_{\text{selec}}$ and ACC$_{\text{pred}}$, and a positive value indicates a high possibility of co-adaptation. We highlight the **First**, **Second** and **Third** performance for each dataset. ProtoGate achieves well-aligned performance for feature selection and prediction.

| Methods | Syn1$_{(+)}$ | | | Syn2$_{(+)}$ | | | Syn3$_{(-)}$ | | |
|---|---|---|---|---|---|---|---|---|---|
| | F1$_{\text{selec}}$ | ACC$_{\text{pred}}$ | Diff. | F1$_{\text{selec}}$ | ACC$_{\text{pred}}$ | Diff. | F1$_{\text{selec}}$ | ACC$_{\text{pred}}$ | Diff. |
| RF | $0.1461 \pm 0.0367$ | $57.08 \pm 6.48$ | 3 | $0.1921 \pm 0.0230$ | $59.44 \pm 5.24$ | 1 | $0.2232 \pm 0.0241$ | $56.33 \pm 9.08$ | -1 |
| Lasso | $0.0905 \pm 0.0197$ | $54.55 \pm 6.14$ | 2 | $0.1130 \pm 0.0070$ | $52.42 \pm 6.69$ | 0 | $0.0900 \pm 0.0179$ | $55.30 \pm 7.44$ | 2 |
| STG | $0.2656 \pm 0.0420$ | $58.65 \pm 9.03$ | -1 | $0.2247 \pm 0.0904$ | $58.28 \pm 8.36$ | -2 | $0.2846 \pm 0.1802$ | $54.00 \pm 9.09$ | -7 |
| TabNet | $0.0843 \pm 0.0172$ | $48.59 \pm 6.55$ | 1 | $0.0642 \pm 0.0246$ | $49.57 \pm 5.38$ | 0 | $0.0605 \pm 0.0200$ | $48.45 \pm 8.31$ | 0 |
| L2X | $0.1599 \pm 0.0710$ | $52.89 \pm 7.51$ | -3 | $0.1873 \pm 0.0976$ | $55.78 \pm 6.97$ | -1 | $0.0984 \pm 0.0889$ | $55.92 \pm 7.30$ | 2 |
| INVASE | $0.1763 \pm 0.0456$ | $55.36 \pm 9.00$ | -1 | $0.1553 \pm 0.0338$ | $60.28 \pm 8.61$ | 6 | $0.1332 \pm 0.0265$ | $58.75 \pm 8.70$ | 5 |
| REAL-X | $0.1850 \pm 0.0438$ | $47.54 \pm 9.51$ | -7 | $0.2328 \pm 0.0729$ | $55.20 \pm 6.38$ | -6 | $0.2630 \pm 0.0567$ | $56.48 \pm 9.34$ | -1 |
| LLSPIN | $0.1060 \pm 0.0246$ | $54.96 \pm 9.49$ | 2 | $0.1692 \pm 0.0795$ | $56.18 \pm 5.80$ | 1 | $0.1031 \pm 0.0635$ | $52.35 \pm 8.32$ | -2 |
| LSPIN | $0.1466 \pm 0.0380$ | $59.04 \pm 9.24$ | 5 | $0.1911 \pm 0.0389$ | $59.40 \pm 8.07$ | 1 | $0.1927 \pm 0.0645$ | $58.09 \pm 6.41$ | 2 |
| ProtoGate | $0.2948 \pm 0.0728$ | $58.68 \pm 6.28$ | -1 | $0.2922 \pm 0.0943$ | $60.67 \pm 8.21$ | 0 | $0.1653 \pm 0.0554$ | $56.16 \pm 6.82$ | 0 |

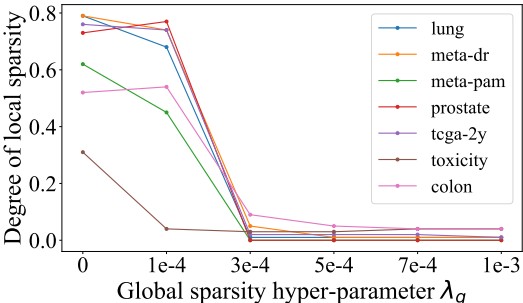

(a) Degree of local sparsity under different $\lambda_g$

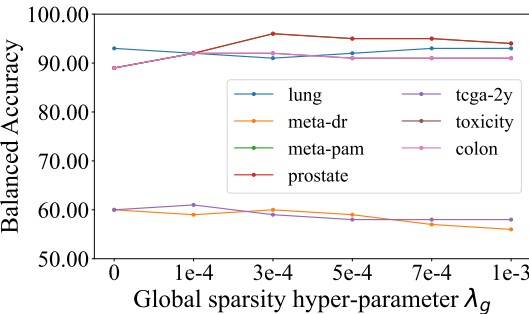

(b) Balanced accuracy under under different $\lambda_g$

*Figure 3.* Comparison of different values of global sparsity hyper-parameter $\lambda_g$ on test set. **(a)** The degree of local sparsity averaged over 25 runs. Increasing $\lambda_g$ reduces the diversity of instance-wise selected features. **(b)** The balanced accuracy averaged over 25 runs. Increasing $\lambda_g$ does not guarantee improvement in accuracy.

another extreme, when $\lambda_g = 0$, the feature selector in Proto-Gate has no explicit restrictions on the homogeneity across samples, and the degree of local sparsity is high. Figure 3(b) further shows that ProtoGate achieves the best test accuracy when selecting features locally ($\mathcal{Q} > 0$), which aligns with the domain knowledge that heterogeneity across samples is important for accurate predictions on biomedical data. The results suggest that ProtoGate achieves outstanding performance by considering both homogeneity and heterogeneity

across samples for feature selection.

### 4.3. Co-adaptation Analysis

We evaluate ProtoGate and benchmark feature selection models on the synthetic datasets to examine their correctness in feature selection and susceptibility to the co-adaptation problem. We use the same experimental settings as real-world datasets and change the range of searching hyper-parameter for each model to achieve their optimal performance. Following (Yang et al., 2022; Jethani et al., 2021), we measure the quality of selected features by computing the F1 score with predicted masks and ground truth masks, and the results are averaged over 25 runs.

**Synthetic datasets.** We generate three synthetic datasets by adapting the nonlinear datasets used in (Yang et al., 2022; Yoon et al., 2018; Jethani et al., 2021), and the exact data models are described in Appendix B.2. Each dataset has 200 samples of 100 features, which is only 10% of the samples and 10 times more features compared to (Yang et al., 2022). All feature values are sampled independently from $\mathcal{N}(0, \mathbf{I})$, where $\mathbf{I}$ is an $100 \times 100$ identity matrix. Each dataset has two classes, and we make the data distribution imbalance by generating 50 and 150 samples for two classes respectively.

We purposely design Syn3$_{(-)}$ to examine the inductive bias in ProtoGate. Note that the absolute value function is an even function. Two samples with opposite values of the same feature are likely to have equal logit values, and then they belong to the same class. However, the opposite values tend to mean a long distance between them, and they should not belong to the same class according to the clustering assumption. Therefore, prototype-based models are expected to perform poorly in this regime. We implement it by adding absolute value function $|x_9|$ in the first class of Syn3$_{(-)}$ to observe the performance degradation in ProtoGate. Because

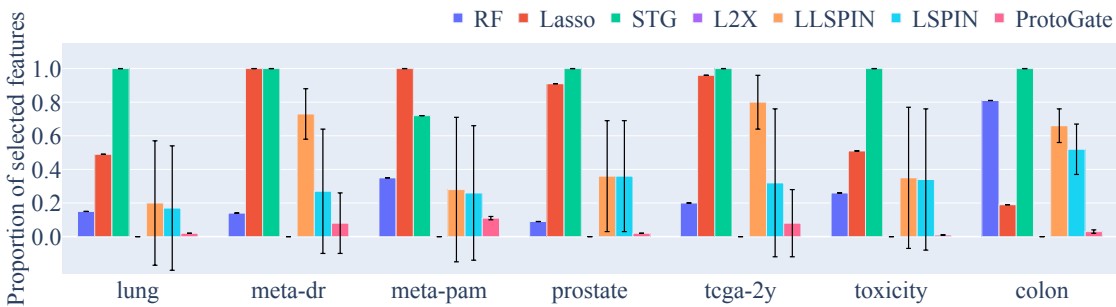

*Figure 4.* Comparison of the feature selection sparsity on real-world datasets. We report the mean ± standard deviation of the proportion of selected features on test samples, averaged over 25 runs. ProtoGate learns sparser patterns than others by a clear margin except for L2X.

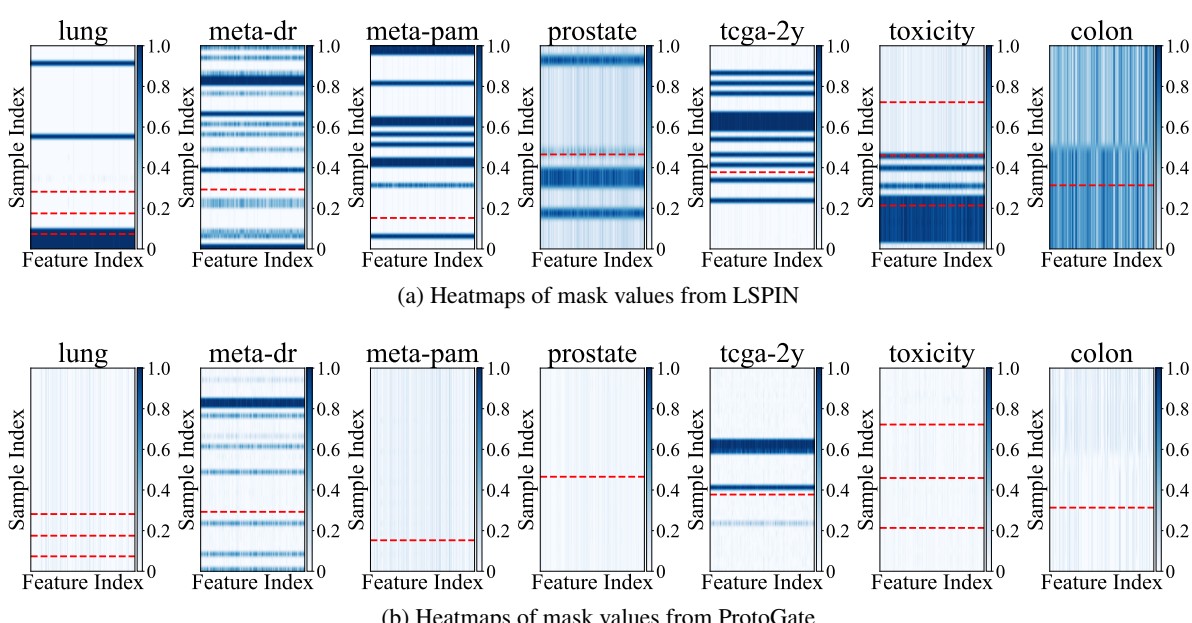

*Figure 5.* Comparison of the selected features on real-world datasets. We plot the heatmaps of predicted mask values $s^{(i)}$ of test samples, where the x-axis refers to the indices of features, and the y-axis refers to the indices of samples. We align the heatmaps of different datasets by adjusting the aspect ratio. The samples are sorted according to their ground truth labels, and the red dash lines separate samples of different classes. Observing the heatmaps, we find that ProtoGate consistently learns more homogeneous feature patterns across samples compared to LSPIN across various real-world datasets. Note that the different feature selection results across samples demonstrate that ProtoGate preserves the ability to identify and leverage heterogeneous feature patterns for local feature selection.

of the evenness of absolute value, two samples with opposite values of $x_9$ are likely to be of the same class, which is against the clustering assumption.

**Results.** On $\text{Syn1}_{(+)}$ and $\text{Syn2}_{(+)}$, ProtoGate achieves better or comparable performance in feature selection and classification than benchmark methods. On $\text{Syn3}_{(-)}$, ProtoGate performs poorly as expected. Although $\text{Syn1}_{(+)}$ and $\text{Syn2}_{(+)}$ also contain even functions like square and absolute value, they also have many other informative features that do not utilise the even functions to compute logit value. Therefore, the side effect of even functions is diluted in $\text{Syn1}_{(+)}$ and $\text{Syn2}_{(+)}$.

We also find the LSPIN exhibits visible misalignment in feature selection and prediction. On $\text{Syn1}_{(+)}$, LSPIN achieves the best classification accuracy, but the quality of selected features is much worse, with a rank of six out of ten methods. In other words, LSPIN simply overfits the dataset without correctly identifying the informative features, denoting a severe co-adaptation problem. In contrast, ProtoGate has consistently non-positive rank differences between $\text{F1}_{\text{selec}}$ and $\text{ACC}_{\text{pred}}$, showing that the co-adaptation does not occur. The results demonstrate that ProtoGate can achieve a well-aligned performance of feature selection and classification, guaranteeing the quality of selected features.

# 5. Conclusion

We present ProtoGate, a prototype-based neural model for local feature selection on high-dimensional and low-sample-size datasets. ProtoGate selects features in a global-to-local manner and makes predictions with an interpretable prototype-based model. The experimental results on real-world datasets demonstrate that ProtoGate improves classification accuracy and interpretability by attending to both homogeneity and heterogeneity across samples. The analysis of synthetic datasets further reveals that ProtoGate can effectively avoid the co-adaptation problem by utilising a prototype-based predictor without learnable parameters. Although we evaluate ProtoGate only on classification tasks in this paper, it is readily extendable and applicable to other biomedical tasks, including regression.

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

# Appendix for submission "ProtoGate: Prototype-based Neural Networks with Local Feature Selection for Tabular Biomedical Data"

## A. Model Design

### A.1. Algorithm for Training ProtoGate

---

**Algorithm 1** Training Procedure of ProtoGate

---

**Input:** training samples $X \in \mathbb{R}^{N \times D}$, ground truth labels $Y \in \mathbb{R}^N$, global-to-local feature selector $S_{\mathbf{W}}$, prototype-based classifier $F$, sparsity hyper-parameters $(\lambda_g, \lambda_l)$, number of nearest neighbours $K$, total training epochs $E$, learning rate $\alpha$
**Output:** trained model $S_{\mathbf{W}}$, prototype base $\mathcal{B}$
  $\mathbf{W} \leftarrow$ GaussianInitialisation() {Initialise the weights of feature selector}
  **for** $e \leftarrow 1$ to $E$ **do**
    $\mathcal{B} \leftarrow \{\}$ {Initialise the prototype base as an empty set}
    **for** $i \leftarrow 1$ to $N$ **do**
      $\boldsymbol{x}_{\text{masked}}^{(i)} \leftarrow \boldsymbol{x}^{(i)} \odot S_{\mathbf{W}}(\boldsymbol{x}^{(i)})$ {Select instance-wise features for training samples}
      $\mathcal{B} \leftarrow \mathcal{B} \cup \{(\boldsymbol{x}_{\text{masked}}^{(i)}, y^{(i)})\}$ {Add masked samples and their labels to the prototype base}
    **end for**
    **for** $i \leftarrow 1$ to $N$ **do**
      $\boldsymbol{x}_{\text{masked}}^{(i)} \leftarrow \boldsymbol{x}^{(i)} \odot S_{\mathbf{W}}(\boldsymbol{x}^{(i)})$ {Select instance-wise features for training samples}
      $P_{\mathcal{B}}^{(i)} \leftarrow$ NeuralSort($\mathcal{B}, \boldsymbol{x}_{\text{masked}}^{(i)}$) {Compute the permutation matrix for $\boldsymbol{x}_{\text{masked}}^{(i)}$}
      $\hat{y}^{(i)} \leftarrow F(\mathcal{B}, \boldsymbol{x}_{\text{masked}}^{(i)}, K)$ {Classify the query sample with $K$ nearest prototypes}
    **end for**
    $L = \frac{1}{N} \sum_{i=1}^{N} \left( \ell_{\text{pred}}(P_{\mathcal{B}}^{(i)}, \boldsymbol{x}^{(i)}, y^{(i)}) + R(\mathbf{W}^{[1]}, \boldsymbol{s}^{(i)}, \lambda_g, \lambda_l) \right)$ {Compute the training loss}
    $\mathbf{W} \leftarrow \mathbf{W} - \alpha \nabla_{\mathbf{W}} L$ {Update the weights of feature selector}
  **end for**
  return $S_{\mathbf{W}}, \mathcal{B}$

---

### A.2. Computation of the Regularisation Term

In line with the approaches presented in (Yamada et al., 2020; Yang et al., 2022), we employ the expectation of the $\ell_0$ norm to ensure differentiability. Note that the mask value $\boldsymbol{s}^{(i)}$ is obtained with the network output $\boldsymbol{\mu}^{(i)}$ and injected noise $\boldsymbol{\epsilon}^{(i)}$. Consequently, we can utilise standard optimization algorithms such as stochastic gradient descent to update the learnable parameters within the global-to-local feature selector in ProtoGate. The regularisation term can be computed as follows:

$$
\begin{aligned}
R(\mathbf{W}^{[1]}, \boldsymbol{s}^{(i)}, \lambda_g, \lambda_l) &= \lambda_g ||\mathbf{W}^{[1]}||_1 + \mathbb{E}\left[\lambda_l ||\boldsymbol{s}^{(i)}||_0\right] \\
&= \lambda_g ||\mathbf{W}^{[1]}||_1 + \lambda_l \sum_{d=1}^{D} \mathbb{P}(\mu_d^{(i)} + \epsilon_d^{(i)} > 0) \\
&= \lambda_g ||\mathbf{W}^{[1]}||_1 + \lambda_l \sum_{d=1}^{D} \left[1 - \mathbb{P}(\mu_d^{(i)} + \epsilon_d^{(i)} \leq 0)\right] \\
&= \lambda_g ||\mathbf{W}^{[1]}||_1 + \lambda_l \sum_{d=1}^{D} \left[1 - \Phi\left(\frac{-\mu_d^{(i)}}{\sigma}\right)\right] \\
&= \lambda_g ||\mathbf{W}^{[1]}||_1 + \lambda_l \sum_{d=1}^{D} \Phi\left(\frac{\mu_d^{(i)}}{\sigma}\right) \\
&= \lambda_g ||\mathbf{W}^{[1]}||_1 + \lambda_l \sum_{d=1}^{D} \left(\frac{1}{2} - \frac{1}{2}\text{erf}(-\frac{\mu_d^{(i)}}{\sqrt{2}\sigma})\right)
\end{aligned}
\tag{7}
$$

## B. Reproducibility

### B.1. Real-word Datasets

Table 3. Details of seven real-world tabular biomedical datasets.

| Dataset | # Samples | # Features | # Classes | # Samples per class |
|---------|-----------|------------|-----------|---------------------|
| lung | 197 | 3312 | 4 | [139, 21, 20, 17] |
| meta-dr | 200 | 4160 | 2 | [139, 61] |
| meta-pam | 200 | 4160 | 2 | [167, 33] |
| prostate | 102 | 5966 | 2 | [52, 50] |
| tcga-2y | 200 | 4381 | 2 | [122, 78] |
| toxicity | 171 | 5748 | 4 | [45, 45, 42, 39] |
| colon | 62 | 2000 | 2 | [40, 22] |

All datasets are publicly available, and the details are listed in Table 3. Four datasets are available online (https://jundongl.github.io/scikit-feature/datasets): **lung** (Bhattacharjee et al., 2001), **Prostate-GE** (referred to as 'prostate') (Singh et al., 2002), **TOX-171** (referred to as 'toxicity') (Bajwa et al., 2016) and **colon** (Ding & Peng, 2005).

In accordance with the methodology presented in (Margeloiu et al., 2023), we derived two datasets from the **METABRIC** dataset (Curtis et al., 2012). We combined the molecular data with the clinical label 'DR' to create the **'meta-dr'** dataset, and we combined the molecular data with the clinical label 'Pam50Subtype' to create the **'meta-pam'** dataset. Because the label 'Pam50Subtype' was very imbalanced, we transformed the task into a binary task of basal vs non-basal by combining the classes 'LumA', 'LumB', 'Her2', 'Normal' into one class and using the remaining class 'Basal' as the second class. For both 'meta-dr' and 'meta-pam', we selected the Hallmark gene set (Liberzon et al., 2015) associated with breast cancer, and the new datasets contain 4160 expressions (features) for each patient. We randomly sampled **200** patients while maintaining stratification to create the final datasets, as our focus is on the HDLSS regime.

Following (Margeloiu et al., 2023), we also derived **'tcga-2y'** dataset from the **TCGA** dataset (Tomczak et al., 2015). We combined the molecular data and the label 'X2yr.RF.Surv' to create the **'tcga-2ysurvival'** dataset. Similar to the previous datasets, we selected the Hallmark gene set (Liberzon et al., 2015) associated with breast cancer, resulting in 4381 expressions (features). We randomly sampled **200** patients while maintaining stratification to create the final datasets, as our focus is on the HDLSS regime.

### B.2. Synthetic Datasets

The synthetic datasets are adapted from the nonlinear datasets in (Yoon et al., 2018; Yang et al., 2022). Specifically, we generate three synthetic datasets: $\text{Syn1}_{(+)}$, $\text{Syn2}_{(+)}$, and $\text{Syn3}_{(-)}$, which are designed for the classification task. Each sample is characterized by 100 features, where the feature values are independently sampled from a Gaussian distribution $\mathcal{N}(0, \mathbf{I})$, with $\mathbf{I}$ representing a $100 \times 100$ identity matrix. The ground truth label (target) $y$ for each sample is computed by

$$y = \mathbb{1}\left(\frac{1}{1 + \text{logit}(\boldsymbol{x})} > 0.5\right) \tag{8}$$

where $\mathbb{1}(\cdot)$ is the indicator function. For each samples, the $\text{logit}(\boldsymbol{x})$ is computed with a small proportion of its features:

$$\textbf{Syn1}_{(+)}\textbf{:} \ \text{logit} = \begin{cases} \exp(x_1 x_2 - x_3) & if \ x_{11} < 0 \\ \exp(x_3^2 + x_4^2 + x_5^2 + x_6^2 - 4) & \text{otherwise} \end{cases} \tag{9}$$

$$\textbf{Syn2}_{(+)}\textbf{:} \ \text{logit} = \begin{cases} \exp(x_3^2 + x_4^2 + x_5^2 + x_6^2 + x_7^2 - 4) & if \ x_{11} < 0 \\ \exp(-10 \sin(0.2x_7) + |x_8| + x_9^2 + \exp(-x_{10}) - 2.4) & \text{otherwise} \end{cases} \tag{10}$$

$$\textbf{Syn3}_{(-)}\textbf{:} \ \text{logit} = \begin{cases} \exp(x_1 x_2 + |x_9|) & if \ x_{11} < 0 \\ \exp(-10 \sin(0.2x_7) + |x_8| + x_9^2 + \exp(-x_{10}) - 2.4) & \text{otherwise} \end{cases} \tag{11}$$

Within each dataset, the two classes have a minimum of two informative features in common. For example, in $\text{Syn1}_{(+)}$, both class one and class two share $(x_3, x_{11})$ as the informative features. To introduce class imbalance, we intentionally generate 150 samples for class one and 50 samples for class two. Note that we purposely design $\text{Syn3}_{(-)}$ to examine the clustering assumption in ProtoGate by adding even function $|x_9|$. Section 4.3 further discusses the rationale behind this choice.

Compared to previous studies (Yang et al., 2022; Yoon et al., 2018), we aim to enhance the difficulty of the synthetic datasets by considering four key aspects. Firstly, we only generate 200 samples for each dataset, which is only 10% of the samples in (Yang et al., 2022). Secondly, each sample has 100 features, which is ten times more than that in (Yang et al., 2022). Thirdly, our synthetic datasets are imbalanced. Lastly, we incorporate a greater number of overlapping informative features between the two classes.

### B.3. Data Preprocessing

Following the methodology presented in (Margeloiu et al., 2023), we perform Z-score normalization on each dataset prior to training the models. This normalization process involves two steps. First, we compute the mean and standard deviation of each feature in the training data. Using these statistics, we transform the training samples to have a mean of zero and a variance of one for each feature. Subsequently, we apply the same transformation to the validation and test data before conducting evaluations.

### B.4. Computing Resources

We trained over 15,000 models (including over 3,000 of ProtoGate) for evaluations. All the experiments were conducted on a machine equipped with an NVIDIA A100 GPU with 40GB memory and an Intel(R) Xeon(R) CPU (at 2.20GHz) with six cores. The operating system used was Ubuntu 20.04.5 LTS.

### B.5. Training Details and Hyper-parameter Tuning

**Software implementation.** We implemented ProtoGate with Pytorch Lightning (Falcon & The PyTorch Lightning team, 2019): the global-to-local feature selector is implemented from scratch, and the DKNN predictor is adapted from its official implementation (`https://github.com/ermongroup/neuralsort`). Note that we optimised the speed of the official implementation of DKNN with matrix operators in PyTorch (Paszke et al., 2019). We re-implemented LSPIN/LLSPIN because the official implementation (`https://github.com/jcyang34/lspin`) used a different evaluation setup from ours: we report the mean $\pm$ standard deviation number of selected features, while they report the median number of selected features. We implemented LightGBM using its open-source implementation (`https://github.com/microsoft/LightGBM`). With scikit-learn (Pedregosa et al., 2011), Random Forest (`https://scikit-learn.org/stable/modules/generated/sklearn.ensemble.RandomForestClassifier`), KNN (`https://scikit-learn.org/stable/modules/generated/sklearn.neighbors.KNeighborsClassifier`) and Lasso (`https://scikit-learn.org/stable/modules/generated/sklearn.linear_model.Lasso`). For other benchmark methods, we used their open-source implementations: STG (`https://github.com/runopti/stg`), TabNet (`https://github.com/dreamquark-ai/tabnet`), L2X (`https://github.com/Jianbo-Lab/L2X`), INVASE (`https://github.com/vanderschaarlab/mlforhealthlabpub/tree/main/alg/invase`) and REAL-X (`https://github.com/rajesh-lab/realx`).

We implemented a uniform pipeline using PyTorch Lightning to ensure consistency and reproducibility. We further fixed the random seeds for data loading and evaluation throughout the training and evaluation process. This ensured that ProtoGate and all benchmark models were trained and evaluated on the same set of samples.

Note that all the libraries utilised in this study adhere to open-source licenses. Specifically, the scikit-learn and the INVASE implementation follow the BSD-3-Clause license, Pytorch Lightning follows the Apache-2.0 license, and the others follow the MIT license.

**Training procedures.** In this section, we outline the key training settings for ProtoGate and all benchmark methods. We made diligent efforts to ensure a fair comparison among the benchmark methods whenever possible. For example, we employed the same predictor architecture in LSPIN, MLP, and STG, as these models share similar design principles.

- **ProtoGate** has a three-layer feature selector. The number of neurons in the hidden layer is 200 for real-world datasets

and 100 for synthetic datasets. And the activation function is $tanh$ for all layers. The model is trained for 10,000 iterations using early stopping with patience 500 on the validation loss. We used the suggested temperature parameter $\tau = 16.0$ in NeuralSort (Grover et al., 2019).

- **LSPIN, LLSPIN and STG** have a feature selector with the same architecture as that in ProtoGate. For LSPIN/STG, the predictor is a feed-forward neural network with hidden layers of $[100, 100, 10]$ with $tanh$ activation function. And we used the same architecture of predictor for **MLP**. For LLSPIN, the architecture of the predictor is the same, but the activation functions are removed. The standard deviation $\sigma$ for injected noise is $0.5$. The model is trained for 7,000 iterations using early stopping with patience 500 on the validation loss.

- **TabNet** has a width of eight for the decision prediction layer and the attention embedding for each mask and 1.5 for the coefficient for feature reusage in the masks. The model is trained with Adam optimiser with a momentum of 0.3 and gradient clipping at 2.

- **L2X, INVASE and REAL-X** have the default architecture as published (Chen et al., 2018; Yoon et al., 2018; Jethani et al., 2021). The feature selector network has two hidden layers of $[100, 100]$, and the predictor network has two hidden layers of $[200, 200]$. They all use the $relu$ activation after layers. For convergence and computation efficiency, L2X is trained for 7,000 iterations, INVASE is trained for 5,000 epochs and REAL-X is trained for 1,000 iterations.

- **LightGBM** has 200 estimators, feature bagging with 30% of the features, a minimum of two instances in a leaf. It is trained for 10,000 iterations to minimise the weighted cross-entropy loss using early stopping with patience 100 on the validation loss.

- **Random Forest** has 500 estimators, feature bagging with the square root of the number of features, and used balanced weights from class distribution.

- **KNN** measured the distance between samples with Euclidean distance and used uniform weights to compute the majority class in the neighbourhood.

- **Lasso** is trained for 10,000 iterations to minimise the weighted loss with the SAGA solver, and the tolerance for early stopping is set as $1e - 4$.

**Hyper-parameter tuning.** To ensure optimal performance, we initially identified a suitable range of hyperparameters for each model to facilitate convergence. Subsequently, we conducted a grid search within this predefined range to determine the optimal hyperparameter settings. The selection of models was based on their balanced accuracy on the validation sets averaged over 25 runs. It is worth noting that tuning hyperparameters in LSPIN can be challenging, particularly for real-world datasets. Therefore, we followed the recommendations in the original paper (Yang et al., 2022) and employed Optuna (Akiba et al., 2019) to fine-tune the hyperparameters for LSPIN.

Table 4 lists the searching range of hyper-parameters in ProtoGate, and Table 5 lists the searching range of hyper-parameters in feature selection benchmark methods. Following (Yang et al., 2022; Margeloiu et al., 2023), we performed hyper-parameter searching for other methods within the same ranges for real-world and synthetic datasets. For **MLP**, we used Optuna to find the optimal learning rate within $[1e - 3, 1e - 1]$. For **LightGBM**, we performed a grid search for the learning rate in $\{1e - 2, 1e - 1\}$ and maximum depth in $\{1, 2\}$. For **Random Forest**, we performed a grid search for the maximum depth in $\{3, 5, 7\}$ and the minimum number of instances in a leaf in $\{2, 3\}$. For **KNN**, we performed a grid search of the number of nearest neighbours in $\{1, 3, 5\}$. For **Lasso**, we performed a grid search of the regularisation strength in $\{1, 1e1, 1e2, 1e3\}$.

*Table 4.* Searching range of hyper-parameters in ProtoGate.

| Datasets | Global Sparsity $\lambda_g$ | Local Sparsity $\lambda_l$ | $K$ | Learning Rate $\alpha$ |
|---|---|---|---|---|
| Real-word | $\{1e - 3\}$ | $\{1e - 4, 2e - 4, 3e - 4, 4e - 4, 6e - 4\}$ | $\{1, 2, 3, 4, 5\}$ | $\{5e - 2, 7.5e - 2, 1e - 1\}$ |
| Synthetic | $\{1e - 2, 1.5e - 2, 2e - 2\}$ | $\{0, 1e - 4, 3e - 4\}$ | $\{3\}$ | $\{1e - 1\}$ |

**Training considerations.** ProtoGate can require large training overhead, mostly for tuning the hyper-parameters compared to some existing models, since we need to consider the interplay between $\lambda_g$, $\lambda_l$ and $K$. ProtoGate also stores all training samples in the prototype base $\mathcal{B}$, leading to higher memory consumption on large datasets than benchmark methods. Because we mainly focus on the HDLSS datasets, memory consumption is not a major problem in this regime.

*Table 5.* Searching ranges of hyper-parameters for feature selection benchmark methods. Note that the ranges for LSPIN/LLSPIN on real-world datasets are intervals instead of sets because we used Optuna to search for the optimal hyper-parameter settings.

| Datasets | Methods | $\lambda$ for sparsity | Learning Rate |
|---|---|---|---|
| Real-world | TabNet | $\{1e-4, 1e-3, 1e-2, 1e-1\}$ | $\{1e-2, 2e-2, 3e-2\}$ |
| | STG | $\{35, 40, 45, 50, 55\}$ | $\{3e-3\}$ |
| | L2X | $\{1, 5, 10\}$ | $\{1e-4\}$ |
| | INVASE | $\{1, 1.5, 2\}$ | $\{1e-4\}$ |
| | REAL-X | $\{1, 5, 10, 30, 50\}$ | $\{1e-4\}$ |
| | LSPIN/LLSPIN | $[5e-4, 1.5e-3]$ | $[5e-2, 1e-1]$ |
| Synthetic | TabNet | $\{1e-2, 1e-1, 5e-1\}$ | $\{1e-2\}$ |
| | STG | $\{1, 3, 5\}$ | $\{1e-1\}$ |
| | L2X | $\{1, 5, 10\}$ | $\{1e-4\}$ |
| | INVASE | $\{1, 1.5, 2\}$ | $\{1e-4\}$ |
| | REAL-X | $\{1, 5, 10, 30, 50\}$ | $\{1e-4\}$ |
| | LSPIN/LLSPIN | $\{1e-2, 5e-2, 1e-1\}$ | $\{1e-1\}$ |

## C. Ablation Studies on Prototype-based Predictor

### C.1. Impact of the DKNN Predictor

We now investigate how the prototype-based predictor impacts classification performance. For a fair comparison, we replace the DKNN predictor with a linear head network or an MLP, and then tune the hyper-parameter for global sparsity $\lambda_g$ by searching within $\{1e-4, 2e-4, 3e-4\}$.

As shown in Table 6, the DKNN predictor consistently outperforms other predictors. We attribute the performance improvement to the appropriate inductive bias in prototype-based classification and the reduction in learnable parameters. In ProtoGate, only the feature selector needs training, while other local feature selection methods have learnable predictors with vast amounts of parameters to optimise. We also find that simply combining a global-to-local feature selector and an MLP/linear prediction head does not outperform LSPIN/LLSPIN. This further indicates that a prototype-based predictor is the key to the high accuracy of ProtoGate.

*Table 6.* Balanced accuracy for different predictors on test samples, averaged over 25 runs. We **bold** the highest accuracy for each dataset. The prototype-based classifier consistently outperforms linear and MLP predictors on all datasets.

| Predictors | lung | meta-dr | meta-pam | prostate | tcga-2y | toxicity | colon |
|---|---|---|---|---|---|---|---|
| MLP | $69.97 \pm 9.17$ | $56.00 \pm 6.37$ | $93.62 \pm 6.04$ | $89.13 \pm 6.36$ | $54.74 \pm 8.11$ | $90.36 \pm 5.61$ | $80.95 \pm 7.77$ |
| Linear Head | $66.51 \pm 12.45$ | $56.10 \pm 8.95$ | $93.20 \pm 6.18$ | $89.87 \pm 5.80$ | $56.60 \pm 8.20$ | $90.29 \pm 5.93$ | $79.45 \pm 6.23$ |
| **DKNN** | $\mathbf{93.44 \pm 6.37}$ | $\mathbf{60.43 \pm 7.61}$ | $\mathbf{95.96 \pm 3.93}$ | $\mathbf{90.58 \pm 5.64}$ | $\mathbf{61.18 \pm 6.47}$ | $\mathbf{92.34 \pm 5.67}$ | $\mathbf{81.10 \pm 12.14}$ |

### C.2. Ablation Impact of the Number of Nearest Neighbours $K$

In order to evaluate the behaviour of the prototype-based predictor, we conducted experiments using different numbers of nearest neighbours denoted as $K$. Considering the limited sample sizes of the datasets under investigation, we set the maximum number of nearest samples to $K = 5$. All other experimental settings were kept consistent to ensure a fair comparison.

Table 7 presents the results of the ablation experiments on the number of nearest neighbours, demonstrating that the optimal value of $K$ varies across different datasets. It is observed that using a small value of $K$ can make the predictions more sensitive to noise and outliers, resulting in lower accuracy. Notably, ProtoGate consistently achieves high accuracy across the range of $K \in \{3, 4, 5\}$. This finding supports the validity of the clustering assumption for the utilised real-world datasets, as ProtoGate exhibits stable and accurate performance.

*Table 7.* Balanced accuracy on test samples for different numbers of nearest neighbours in prototype-based classification, averaged over 25 runs. We **bold** the highest accuracy for each dataset. A small $K \in \{1, 2\}$ can lead to sensitivity to noise, and the model performs stably with $K \in \{3, 4, 5\}$.

|  | lung | meta-dr | meta-pam | prostate | tcga-2y | toxicity | colon |
|---|---|---|---|---|---|---|---|
| $K=1$ | $87.53 \pm 7.28$ | $50.50 \pm 6.21$ | $73.02 \pm 10.90$ | $75.91 \pm 10.21$ | $57.46 \pm 6.85$ | $75.85 \pm 7.02$ | $70.40 \pm 14.45$ |
| $K=2$ | $92.30 \pm 7.28$ | $56.06 \pm 7.29$ | $90.28 \pm 6.01$ | $86.93 \pm 7.33$ | $59.40 \pm 6.24$ | $88.81 \pm 7.01$ | $77.35 \pm 13.46$ |
| $K=3$ | $\mathbf{93.44 \pm 6.37}$ | $57.82 \pm 8.93$ | $\mathbf{95.96 \pm 3.93}$ | $89.53 \pm 5.64$ | $\mathbf{61.18 \pm 6.47}$ | $91.14 \pm 5.19$ | $\mathbf{81.10 \pm 12.14}$ |
| $K=4$ | $90.34 \pm 7.01$ | $\mathbf{60.43 \pm 7.61}$ | $95.03 \pm 4.77$ | $88.85 \pm 5.87$ | $60.97 \pm 5.60$ | $91.10 \pm 4.93$ | $75.25 \pm 13.34$ |
| $K=5$ | $91.12 \pm 6.36$ | $59.23 \pm 6.88$ | $95.83 \pm 5.89$ | $\mathbf{90.58 \pm 5.64}$ | $60.84 \pm 5.88$ | $\mathbf{92.34 \pm 5.67}$ | $77.50 \pm 8.67$ |

# D. Comparison of Training Duration

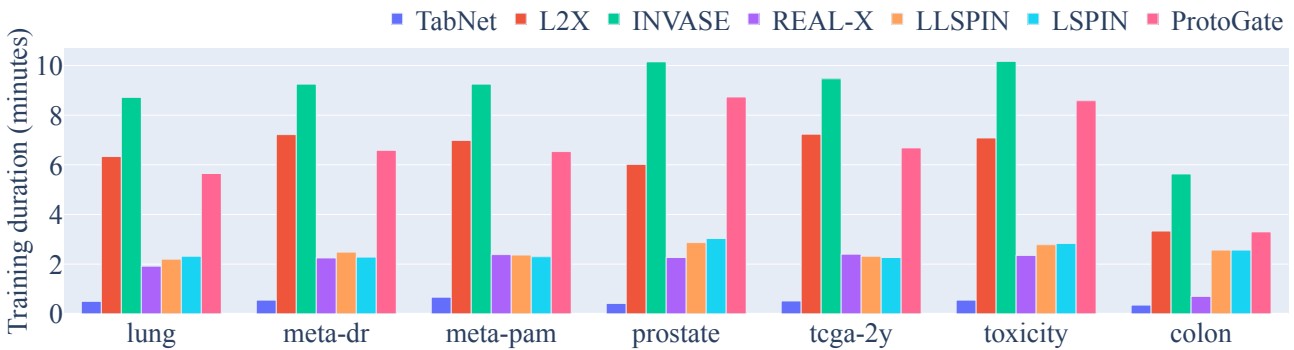

*Figure 6.* Comparison of training duration on real-world datasets. TabNet is the fastest method, but its accuracy is substantially lower than other methods. ProtoGate has a shorter training duration than INVASE and similar computation efficiency as L2X.

# E. Complete Results on the Sparsity of Feature Selection

## E.1. Numerical Results

*Table 8.* Quantitative comparison of the feature selection sparsity on real-world datasets. We report the mean $\pm$ standard deviation of the number of selected features on test samples, averaged over 25 runs. ProtoGate learns sparser patterns than other local feature selection methods by a clear margin except for L2X.

| Methods | lung | meta-dr | meta-pam | prostate | tcga-2y | toxicity | colon |
|---|---|---|---|---|---|---|---|
| RF | $504.76 \pm 0.00$ | $577.60 \pm 0.00$ | $1439.20 \pm 0.00$ | $510.72 \pm 0.00$ | $887.12 \pm 0.00$ | $1507.44 \pm 0.00$ | $1629.72 \pm 0.00$ |
| Lasso | $1618.08 \pm 0.00$ | $4159.92 \pm 0.00$ | $4159.40 \pm 0.00$ | $5434.68 \pm 0.00$ | $4214.56 \pm 0.00$ | $2951.28 \pm 0.00$ | $371.40 \pm 0.00$ |
| STG | $3312.00 \pm 0.00$ | $4157.96 \pm 0.00$ | $2992.00 \pm 0.00$ | $5966.00 \pm 0.00$ | $4381.00 \pm 0.00$ | $5748.00 \pm 0.00$ | $2000.00 \pm 0.00$ |
| L2X | $1.00 \pm 0.00$ | $5.00 \pm 0.00$ | $10.00 \pm 0.00$ | $10.00 \pm 0.00$ | $5.00 \pm 0.00$ | $5.00 \pm 0.00$ | $5.00 \pm 0.00$ |
| LLSPIN | $673.27 \pm 1212.20$ | $3026.77 \pm 642.02$ | $1180.08 \pm 1769.59$ | $2151.05 \pm 1954.80$ | $3486.77 \pm 696.29$ | $1999.12 \pm 2398.65$ | $1311.86 \pm 209.80$ |
| LSPIN | $564.83 \pm 1236.21$ | $1138.51 \pm 1545.96$ | $1073.04 \pm 1661.89$ | $2120.00 \pm 1968.86$ | $1418.35 \pm 1936.41$ | $1979.29 \pm 2387.03$ | $1044.32 \pm 293.67$ |
| ProtoGate | $71.04 \pm 4.96$ | $337.21 \pm 738.81$ | $469.47 \pm 46.90$ | $91.29 \pm 7.20$ | $348.79 \pm 869.23$ | $76.39 \pm 17.42$ | $65.29 \pm 10.69$ |

### E.2. Visualisation of Selected Features on Real-world Datasets

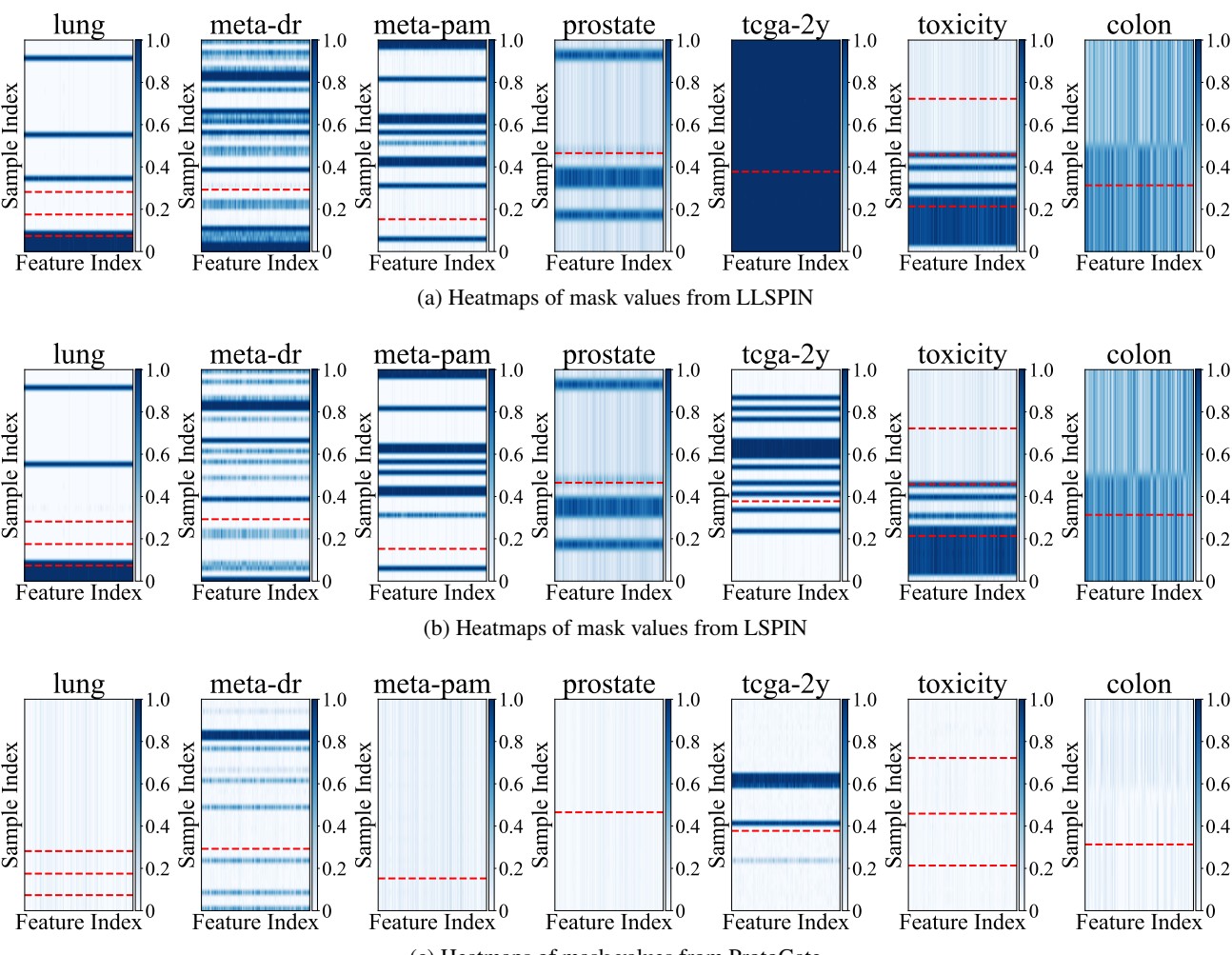

*Figure 7.* Comparison of the mask values on real-world datasets. We plot the heatmaps of predicted mask values $s^{(i)}$ of test samples, where the x-axis refers to the indices of features, and the y-axis refers to the indices of samples[2]. The samples are sorted according to their ground truth labels, and the red dash lines separate samples of different classes. Observing the heatmaps, we find that ProtoGate consistently learns more homogeneous feature patterns across samples compared to LSPIN/LLSPIN across various real-world datasets. Note that the different feature selection results across samples demonstrate that ProtoGate preserves the ability to identify and leverage heterogeneous feature patterns for local feature selection.

---

[2]Different datasets can have different numbers of samples and features, and the number of features should be more than the number of samples in the HDLSS regime. For visualisation purposes, we align them by adjusting the aspect ratio of heatmaps.

