# OpenReview forum: "ProtoGate: Prototype-based Neural Networks with Local Feature Selection for Tabular Biomedical Data"
_ICML.cc/2023/Workshop/IMLH — IMLH 2023 Poster_

### Official Review · Reviewer_Zecc · 2023-06-10
**Significance: The paper addresses the challenges of suboptimal performance and lack of transparency in local feature selection methods for tabular biomedical data. The proposed ProtoGate method shows promising results in terms of predictive performance and feature selection. It outperforms several benchmark methods and provides a more interpretable way of making predictions. The significance of the work lies in its potential to improve the accuracy and interpretability of local feature selection models in the biomedical domain.**

**Rating:** 9
**Confidence:** 3

**Review:**

Quality: The work appears to be of high quality. The authors propose a novel method, ProtoGate, for local feature selection in tabular biomedical data. They provide a detailed description of the method, its design choices, and the rationale behind them. They also conduct experiments on synthetic and real-world datasets to evaluate the performance of ProtoGate and compare it with several benchmark methods. The experiments are well-designed, and the results are presented clearly.

Clarity: The paper is generally well-written and structured. The authors provide clear explanations of the method and the experiments. However, there are some sections where the text could be improved for better clarity. For example, the section on the global-to-local feature selection could be more concise and easier to follow.

Pros:

ProtoGate introduces a unique combination of a prototype-based predictor and a global-to-local feature selector for local feature selection in tabular biomedical data.

The authors conduct comprehensive experiments on synthetic and real-world datasets to evaluate the performance of ProtoGate. They compare it with several benchmark methods and provide statistical analysis of the results.


Cons:

Clarity of some sections: While the paper is generally well-written, there are sections where the text could be improved for better clarity and ease of understanding.
Lack of in-depth discussion: The paper could benefit from a more thorough discussion of the limitations and potential future directions of the proposed method.

---

### Official Review · Reviewer_bfrW · 2023-06-19
**The paper deals with an important problem in biomedical analysis and proposes a novel feature selection method**

**Rating:** 7
**Confidence:** 4

**Review:**

This paper proposes propose ProtoGate, a local feature selection method that incorporates an inductive bias by considering the clustering characteristics of biomedical data. ProtoGate selects features in a global-to-local manner, leading to improved prediction accuracy through the use of a prototype-based model, as demonstrated in experiments on synthetic and real-world datasets.

The idea of selecting instance-wise features with inductive biases from prototype-based model is novel and the paper is well written. Some minor suggestions -

1. The current evaluation datasets are all in small sample size (less than 200 samples). It would be great to implement ProtoGate on relatively larger dataset and benchmark its performance. Also, it would be appreciated if the comparison of running time and computational cost across different methods can be added.

2. It would be desirable if some analysis about the interpretability of the final results can be made (e.g. even some case study to compare how more interpretable of ProtoGate compared to other methods).

---

### Meta-Review · Area_Chair_xdZy · 2023-06-20

**Recommendation:** Accept (Poster)
**Confidence:** 5

**Metareview:**

Both reviewers expressed positive opinions about this paper, appreciating its clarity and extensive experimentation. I kindly ask the authors to carefully consider the identified shortcomings and ensure that these issues are addressed in the final version.

---

### Decision · Program_Chairs · 2023-06-20

Accept (Poster)